# Learning a Domain-Agnostic Policy through Adversarial Representation Matching for Cross-Domain Policy Transfer

## Abstract

The low transferability of learned policies is one of the most critical problems limiting the applicability of learning-based solutions to decision-making tasks. In this paper, we present a way to align latent representations of states and actions between different domains by optimizing an adversarial objective. We train two models, a policy and a domain discriminator, with unpaired trajectories of proxy tasks through behavioral cloning as well as adversarial training. After the latent representations are aligned between domains, a domain-agnostic part of the policy trained with any method in the source domain can be immediately transferred to the target domain in a zero-shot manner. We empirically show that our simple approach achieves comparable performance to the latest methods in zero-shot cross-domain transfer. We also observe that our method performs better than other approaches in transfer between domains with different complexities, whereas other methods fail catastrophically.

## 1 Introduction

Humans have an astonishing ability to learn skills in a highly transferable way. Once we learn the route from home to the station, for example, we can get to the destination using different vehicles (e.g., walking, cycling, or driving) in different environments (e.g., on a map or in the real world) ignoring irrelevant perturbations (e.g., weather, time, or traffic conditions). We find underlying structural similarities between situations, perceive the world, and accumulate knowledge in our way of abstraction. Such abstract knowledge can be readily applicable to various similar situations. This seems easy for humans but not for autonomous agents. Agents trained in reinforcement learning (RL) or imitation learning (IL) often have difficulties in transferring knowledge learned in a specific situation to another. It is because the learned policies are strongly tied to the representation acquired in a specific configuration of training, which is not generalizable even to subtle changes in an agent or an environment.

Previous studies have attempted to address this problem with various approaches. Domain randomization (Tobin et al., 2017; Peng et al., 2018; Andrychowicz et al., 2020) aims to learn a policy robust to environmental changes by having access to multiple training domains, but it cannot handle large domain gaps out of the domain distribution assumed in training such as drastically different observations or agent morphologies. To overcome such domain discrepancies, Gupta et al. (2017) and Liu et al. (2018) proposed methods to learn domain-invariant state representations for imitation, but these methods require paired temporally-aligned datasets across domains and, in addition, need expensive RL steps to adapt to the target domain. More recently, Kim et al. (2020) proposed a method to find cross-domain correspondence of states and actions from unaligned datasets through adversarial training. This method imposes a strong assumption that there is an exact correspondence of states and actions between different domains and learns it as direct mapping functions. The assumption is sometimes problematic when such correspondence is hard to find. For example, if one agent has no leg while another has a few legs, we cannot expect all information on how the agent walks to be translated into another domain.

In this work, we propose a method that does not require the existence of exact correspondence between domains. Our approach learns domain-invariant representations and a common abstract

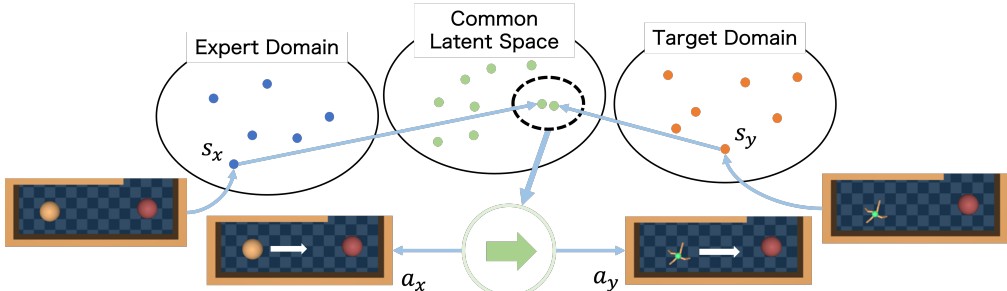

Figure 1: Illustration of a domain-agnostic representation space. Since semantically similar states are close together in the latent space regardless of the original domain, we can transfer knowledge between different domains through the latent space.

policy on them that is shared across different domains (Figure 1). Our model consists of two core components: a policy and a domain discriminator. The policy has three blocks: a state encoder, a common policy, and an action decoder. In the first stage, the model is optimized with an imitation objective and an adversarial objective on a learned state representation simultaneously using an unaligned dataset of proxy tasks. The adversarial training induces the state encoder to generate latent state representations whose domains are indistinguishable to the domain discriminator. Such representations do not contain domain-specific information and thus can work with the common policy. Next, we freeze the parameters of the state encoder and action decoder and only update the common policy in the source domain on the learned feature space to adapt the model to the target task. In this process, as with Kim et al. (2020), we can use any learning algorithm for updating the policy and moreover do not require an expensive RL step interacting with the environment. After the update, combined with the fixed encoder and decoder, the learned common policy can be readily used in either domain in a zero-shot manner.

We conduct experiments on a challenging maze environment (Fu et al., 2020) with various domain shifts: domain shift in observation, action, dynamics, and morphology of an agent. Our experiments show that our approach achieves comparable performance in most settings. We find that our method is effective in the setting of cross-dynamics or cross-robot transfer, where no exact correspondence between domains exists.

In summary, our contributions are as follows:

- We provide a novel method of cross-domain transfer with an unaligned dataset. In contrast to the latest method that learns mappings between domains, our approach aims to acquire a domain-invariant feature space and a common policy on it.

- Our experiments with various domain shifts show that our method achieves comparable performance in transfer within the same agent and better performance than existing methods in cross-dynamics or cross-robot transfer by avoiding direct mapping between domains.

## 2 RELATED WORK

**Cross-Domain Policy Transfer between MDPs** Transferring a learned policy to a different environment is a long-standing challenge in policy learning. Most previous methods acquire some cross-domain metric to optimize and train a policy for a target task using a standard RL algorithm (Gupta et al., 2017; Liu et al., 2018; 2020; Zakka et al., 2022; Fickinger et al., 2022) or a GAIL (Ho & Ermon, 2016)-based approach (Stadie et al., 2017; Franzmeyer et al., 2022). In particular, Franzmeyer et al. (2022) utilize domain confusion as we do in this paper. To calculate the pseudo reward for RL, the distance between temporally-corresponding states (Gupta et al., 2017; Liu et al., 2018) or the distance from the goal (Zakka et al., 2022) in the latent space is often used. Some recent approaches can perform zeros-shot cross-domain transfer as our method does, without interacting with the environment for a target task. Kim et al. (2020); Raychaudhuri et al. (2021) learn mappings between domains while Zhang et al. (2020) impose domain confusion on its state representation for domain shift in observation. Our approach predicts actions without learning cross-domain mappings

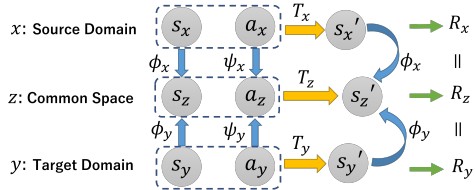

Figure 2: Common latent MDP between a source MDP and a target MDP. The latent MDP is expected to be expressive enough to reproduce the dynamics and the reward of the original MDPs.

and also can handle diverse situations including domain shift in an action space. For cross-robot transfer, Hejna et al. (2020) train a portable high-level policy by using a subgoal position as a cross-robot feature. Gupta et al. (2022) cover morphology distribution to generalize to unseen robots. We intend to perform more direct policy transfer without making domain-specific assumptions.

**State Abstraction for Transfer**   Theoretical aspects of latent state representation have been analyzed in previous studies. There exist several principled methods of state representation learning for transfer such as bisimulation (Castro & Precup, 2010) and successor features (Barreto et al., 2017), although we use neither of them. Recently, Gelada et al. (2019) proved that the quality of the value function is guaranteed if the representation is sufficient to predict the reward and dynamics of the original MDP. In a similar context, Zhang et al. (2020); Sun et al. (2022) provide performance guarantees in multi-task settings or cross-domain transfer without paired relationships between domains.

**Unsupervised Domain Adaptation & Correspondence Learning**   Domain adaptation with unaligned datasets has been intensively studied in computer vision. CycleGAN (Zhu et al., 2017) finds a translation function between domains by generating the corresponding instances in another domain. Similarly to our approach, Tzeng et al. (2017) matches the feature distributions between domains by fooling a domain discriminator and successfully transfers an image classifier across domains. Besides, several studies learn domain-invariant features by optimizing an objective related to temporal relationships of frames in videos (Sermanet et al., 2018; Dwibedi et al., 2019) or cycle-consistency in trajectories of an agent (Zhang et al., 2021; Wang et al., 2022). Such features can be used in the reward shaping for cross-domain imitation through RL (Zakka et al., 2022).

## 3   PROBLEM FORMULATION

We consider a Markov decision process (MDP): $\mathcal{M} = (\mathcal{S}, \mathcal{A}, R, T, \gamma)$, where $\mathcal{S}$ is a state space, $\mathcal{A}$ is an action space, $R : \mathcal{S} \times \mathcal{A} \to \mathbb{R}$ is a reward function, $T : \mathcal{S} \times \mathcal{A} \times \mathcal{S} \to \mathbb{R}_{\geq 0}$ is a transition function, and $\gamma$ is a discount factor. The aim of this paper is to transfer knowledge of a source MDP $\mathcal{M}_x = (\mathcal{S}_x, \mathcal{A}_x, R_x, T_x, \gamma)$ to a target MDP $\mathcal{M}_y = (\mathcal{S}_y, \mathcal{A}_y, R_y, T_y, \gamma)$. Here we assume that these MDPs share a common latent structure which is also an MDP: $\mathcal{M}_z = (\mathcal{S}_z, \mathcal{A}_z, R_z, T_z, \gamma)$. Formally, we assume the existence of state mapping functions $\phi_x : \mathcal{S}_x \to \mathcal{S}_z, \phi_y : \mathcal{S}_y \to \mathcal{S}_z$ and action mapping functions $\psi_x : \mathcal{A}_x \to \mathcal{A}_z, \psi_y : \mathcal{A}_y \to \mathcal{A}_z$ which translate states $s_x, s_y$ or actions $a_x, a_y$ into shared states $s_z$ or actions $a_z$, respectively, satisfying the following relationships:

$$T_z(\phi_x(s_x), \psi_x(a_x)) = \phi_x(T_x(s_x, a_x)), \quad T_z(\phi_y(s_y), \psi_y(a_y)) = \phi_y(T_y(s_y, a_y)),$$
$$R_z(\phi_x(s_x), \psi_x(a_x)) = R_x(s_x, a_x), \quad R_z(\phi_y(s_y), \psi_y(a_y)) = R_y(s_y, a_y).$$

In short, as depicted in Figure 2, we assume that the common latent MDP is expressive enough to reproduce the dynamics and reward structure of the source and target MDP. Our goal is to learn the state mapping functions $\phi_x, \phi_y$ and the action mapping function $\psi_x, \psi_y$ so that any policy learned in the common latent space $\pi_z(a_z|s_z) : \mathcal{S}_z \times \mathcal{A}_z \to \mathbb{R}_{\geq 0}$ can be immediately used in either MDP combined with the obtained mappings. In this paper, we use a deterministic policy, and thus we sometimes denote the latent policy as $\pi_z(s_z) : \mathcal{S}_z \to \mathcal{A}_z$, although we can easily extend it to a stochastic policy. We learn these mappings using state action trajectories of a set of proxy tasks $\mathcal{D}_{d,k} = \{\tau_{d,k,i}\}$, where $\tau_{d,k,i} = \{(s_d^t, a_d^t)\}$ is a successful trajectory of task $k$ in domain $d$, and use the learned relationships in task $k'$ unseen during training. The alignment of the representation is obtained through behavioral cloning (BC) on the trajectories of proxy tasks and adversarial training on the embeddings, which will be described in the following section.

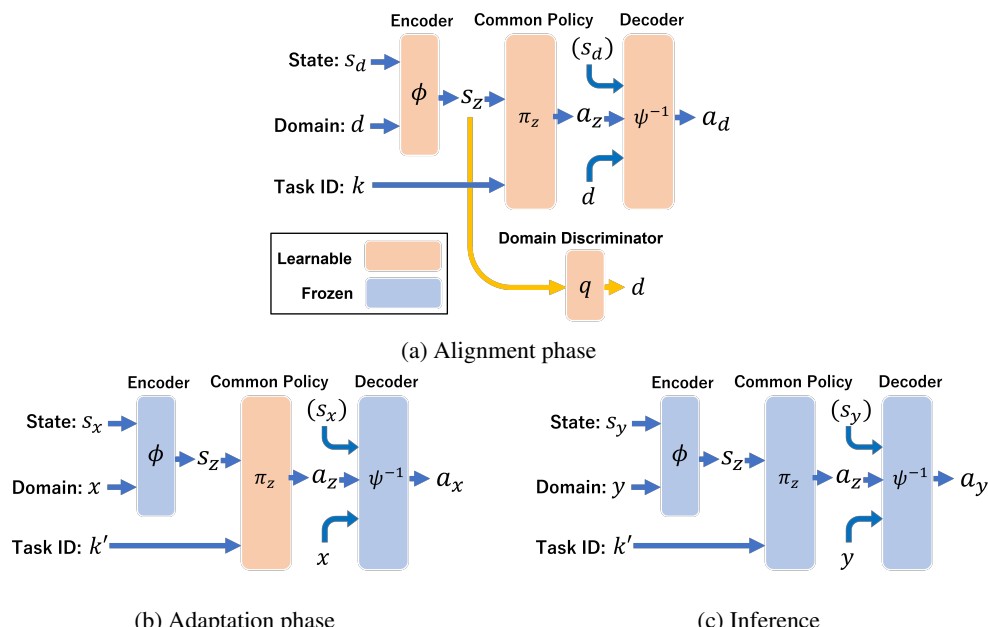

(a) Alignment phase

(b) Adaptation phase            (c) Inference

Figure 3: Overview of the training and inference procedure of our method. (a) In the alignment phase, we jointly train the policy and the discriminator using trajectories of proxy tasks to match the representation between domains. (b) In the adaptation phase, we only update the common policy to adapt to the target task in the source domain. (c) In inference, we can use the updated policy trained in the source domain combined with the encoder and decoder already trained in the alignment phase.

## 4    LEARNING COMMON POLICY THROUGH REPRESENTATION MATCHING

In this work, we aim to learn state mapping functions $\phi_x, \phi_y$, and action mapping functions $\psi_x, \psi_y$ or equivalents, and use them to transfer the policy learned in one domain to another. Our algorithm of cross-domain transfer consists of two steps as illustrated in Figure 3: (i) Cross-domain representation alignment, (ii) Policy adaptation to a target task in the source domain. We call them the *alignment* phase and the *adaptation* phase, respectively. After the adaptation phase, the learned policy of a target task can be immediately used in the target domain without any fine-tuning, additional training interacting with the target domain (Gupta et al., 2017; Liu et al., 2018), or a policy learning in the mapped target domain (Raychaudhuri et al., 2021).

### 4.1    CROSS-DOMAIN REPRESENTATION ALIGNMENT

In the alignment phase, we aim to learn the state and action mappings and acquire a domain-invariant feature space that can be used in either the source domain or the target domain. We represent our policy as a simple feed-forward neural network as shown in Figure 3. It consists of three components: a state encoder, a common policy, and an action decoder. They correspond to $\phi(s)$, $\pi(s_z)$, and $\psi^{-1}(a_z)$, respectively. $\psi^{-1}(a_z)$ can optionally take a raw state $s$ for the action prediction in a domain. Note that we feed domain ID $d$ to the encoder and the decoder instead of using two separate networks for the domains to simplify the architecture and the training. Additionally, we feed task ID $k$ to the policy to deal with multiple proxy tasks simultaneously.

Even when the dataset contains trajectories from both domains performing the same tasks, simply training the policy with such a dataset does not necessarily match the representations of both domains. Figure 4a shows the distribution of the representation of latent state $s_z$ acquired by behavioral cloning using expert trajectories from different domains in our experiment (Maze2D-OA). Although we trained a single policy, the learned representations are completely separated for each domain. We need an additional objective to obtain a better alignment.

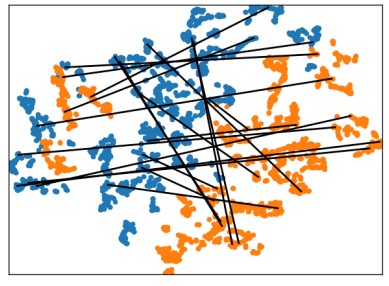 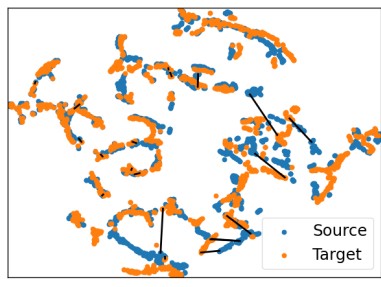

(a) Distribution without adversarial training        (b) Distribution with adversarial training

Figure 4: Distributions of latent state representation $s_z$ with and without adversarial training. These are visualized with t-SNE (Van der Maaten & Hinton, 2008). Here we sample corresponding states from two domains of the Maze2D environment in our experiment and encode them into the learned feature space. Black lines connect 20 corresponding state pairs in each figure. Without adversarial training, the model can learn the policy for different domains on different feature spaces. The adversarial training encourages the model to learn them on the same feature space.

If we have access to temporally-aligned trajectories (i.e. $(\phi_x(s_x^t), \psi_x(a_x^t)) = (\phi_y(s_y^t), \psi_y(a_y^t))$), we can directly learn corresponding representations $\phi_x(s_x), \phi_y(s_y)$ or $\psi_x(a_x), \psi_y(a_y)$ close together as done in Gupta et al. (2017). However, we do not assume such an alignment in the dataset and hence we instead use adversarial training to match the representations. Our purpose is to match the corresponding latent representation of states and actions between domains. In other words, we want the latent representations to contain domain-specific information as little as possible. It can be expressed as mutual information minimization between domains $D$ and latent representation $\mathcal{S}_z$ or $\mathcal{A}_z$. Here we can skip this process on latent actions $a_z$ since $a_z$ is a function of latent states $s_z$. The objective for latent states can be formulated as follows:

$$\min_{\phi} I(D; \mathcal{S}_z) = \min_{\phi} H(D) - H(D|\mathcal{S}_z)$$
$$= \max_{\phi} H(D|\mathcal{S}_z). \tag{1}$$

The second equation holds because we cannot control the domain distribution by the encoder. To optimize this objective, we introduce a domain discriminator $q(d|s_z)$ as an approximation of $p(d|s_z)$ in the objective. The training proceeds in a similar way to that of generative adversarial networks (GAN) (Goodfellow et al., 2014). The discriminator $q(d|s_z)$ predicts which domain each given $s_z$ came from, while the encoder $\phi$ tries to fool the discriminator, as indicated in (1). Combined with the behavioral cloning objective, we have our final objective $\mathcal{L}_{\text{align}}$ for the alignment phase:

$$\min_{\phi, \pi_z, \psi^{-1}} \max_q \mathcal{L}_{\text{align}} = \min_{\phi, \pi_z, \psi^{-1}} \max_q \mathcal{L}_{\text{BC}} + \lambda \mathcal{L}_{\text{adv}}$$
$$\begin{cases} \mathcal{L}_{\text{BC}} = \mathbb{E}_{(s_d, a_d, d, k) \sim \mathcal{D}} \left[ \left( \psi_d^{-1}(\pi_z(\phi_d(s_d), k)) - a_d \right)^2 \right] \\ \mathcal{L}_{\text{adv}} = \mathbb{E}_{(s_d, d) \sim \mathcal{D}} \left[ \log q(d|\phi_d(s_d)) \right], \end{cases} \tag{2}$$

where $\lambda$ is a constant that defines the importance of the adversarial term. More detailed derivation and interpretation can be found in Appendix A. By optimizing the adversarial objective along with imitation, we can obtain an aligned representation as shown in Figure 4b. Intuitively, behavioral cloning shapes the structure of the representation of each domain (Figure 4a), and the adversarial term pulls the entire distributions close together so that they will be identical (Figure 4b). Besides, we also employ task-level alignment by feeding task ID $k$ to the discriminator. We observe that this improves performance in some cases. See Appendix B.3 for the performance comparisons.

## 4.2 POLICY ADAPTATION

In the adaptation phase, we train our policy to successfully perform the target task which we finally would like to solve. We assume that the latent states and the latent actions are already aligned in the alignment phase, and also we have state and action mapping functions between the common latent

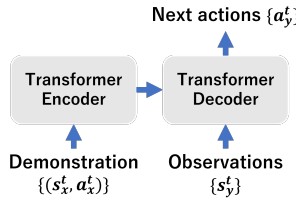

Figure 5: Illustration of a demonstration-conditioned policy. This policy cannot be trained only with data of a source domain.

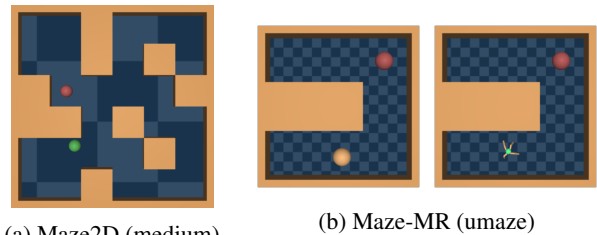

(a) Maze2D (medium)  (b) Maze-MR (umaze)

Figure 6: Pictures of Maze2D and Maze-MR. The red points show the goal, which are not observable for agents.

space and each domain. Thus, our task in this phase is to train the common latent policy for the target task. We can train it solely in the source domain using mapping functions. As described in Figure 3, the weights of the encoder and decoder are fixed during the training so that the common policy is learned on top of the acquired aligned representations. In this phase, we can use any learning algorithm, including reinforcement learning. In our experiments, we update the common policy by behavioral cloning using expert trajectories in the source domain. Therefore, we minimize the following objective in the adaptation phase.

$$\min_{\pi_z} \mathcal{L}_{\text{adapt}} = \min_{\pi_z} \mathcal{L}_{\text{BC}}.$$

This adaptation phase is one of the advantages of using a common policy over a demonstration-conditioned policy (Figure 5) used in previous studies such as Dasari & Gupta (2020). Since a conditional policy cannot be trained with a dataset that only contains demonstrations in the source domain, we cannot update it to adapt to the target task. This constraint requires the model to generalize to demonstrations of an unseen target task in a zero-shot manner, which is difficult especially when the target task is much more advanced than proxy tasks used in the training. We demonstrate the effectiveness of our method from this perspective in our experiment.

## 5 EXPERIMENTS

In the experiment, our aim is to answer these questions: (i) Can our method align the states and actions of a source and a target domain? (ii) Can our method achieve zero-shot cross-domain transfer in various settings? (iii) In which case is our method more effective compared to previous approaches?

### 5.1 ENVIRONMENTS

To evaluate the efficacy of our method, we choose the Maze environment used in the D4RL (Fu et al., 2020) benchmark. It is a multi-task robotic locomotion environment suitable for defining proxy tasks, and its simplicity enables us to evaluate the discrepancy of the alignment we obtain from the ideal one. It also offers multiple agents with different morphologies, such as Point and Ant, with which we can evaluate the performance of cross-domain transfer between significantly different observation and action spaces. Concretely, we use four environments for our evaluation (Two of them are shown in Figure 6). In each environment, agents explore the maze toward a specific goal. The shape of the maze has two variations: *umaze* and *medium*. A task is defined as a 2D position of a goal, and the dataset has expert trajectories of tasks. Note that the starting point can vary within a task. **Maze2D-O** A point agent explores the maze. The observation space has four dimensions (position and velocity for each direction), and the action space has two dimensions. The x-axis and the y-axis are swapped in observations of the target domain. **Maze2D-OA**: In addition to the observation shift of Maze2D-O, actions for both directions are inverted (i.e., multiplied by $-1$). **Maze-D** (Dynamics): In the source domain, inertia is removed and an action directly determines the velocity. Agents in domains share the same observation, but they have different action spaces, dynamics, and state distributions. **Maze-MR** (Multi-Robot): An ant agent is used as a target agent. The observation space and the action space of Ant are 29-dimensional and 8-dimensional, respectively. Transfer between these agents is challenging since they have different morphologies and thus the observation

Table 1: Alignment scores. The values are mean squared errors between the ground-truth states in the source domain and the ones predicted from the corresponding target states. The results are averaged over nine seeds with three fixed goals.

| Task | Ours | Ours-S | GAMA | CCA | IfO |
|---|---|---|---|---|---|
| Maze2D-O (umaze) | $0.11 \pm 0.06$ | $1.98 \pm 0.54$ | $0.09 \pm 0.04$ | $1.99 \pm 2.74$ | $3.60 \pm 1.34$ |
| Maze2D-O (medium) | $0.21 \pm 0.26$ | $4.49 \pm 2.49$ | $0.08 \pm 0.02$ | $0.01 \pm 0.01$ | $4.71 \pm 0.76$ |
| Maze2D-OA (umaze) | $4.50 \pm 1.75$ | $1.72 \pm 0.73$ | $0.11 \pm 0.03$ | $1.99 \pm 2.74$ | $3.60 \pm 1.34$ |
| Maze2D-OA (medium) | $2.26 \pm 1.89$ | $2.88 \pm 0.57$ | $0.10 \pm 0.05$ | $0.01 \pm 0.01$ | $4.71 \pm 0.76$ |

space, action space, and dynamics vary significantly. We also measure the performance in a manipulation environment and show results in Appendix B.6. For other experimental settings, including the detail of datasets, architectures, and the training procedures, refer to Appendix C.1.

## 5.2 BASELINES

For our proposed methods, we used two implementations. In **Ours**, the decoder predicts actions without raw states $s$. In **Ours-S** we feed $s$ to the decoder to compensate domain-specific information lost from latent representations. We compare our methods with the following approaches. **CCA** (Hotelling, 1992) finds invertible linear transformations to the space where unaligned demonstrations are maximally correlated. A target policy is then optimized by reinforcement learning so that the policy can obtain observations similar to the given demonstrations of a target task in the learned space. **IfO** (Gupta et al., 2017) uses dynamic time warping (DTW) (Müller, 2007) to find temporally-corresponding pairs and get the representations of them close together using a context translation model. A target policy is trained in a similar way as CCA using RL. **GAMA** (Kim et al., 2020) is one of the closest approaches to ours. It first learns direct mappings of states and actions between domains in an adversarial way. After that, it updates the source domain policy and then solves the target task by translating the states to those of the source domain, applying the source domain policy, and translating the output actions back to the target domain. The critical difference between this approach and ours is that GAMA uses the source domain policy for the target task, while ours uses a common policy. In addition, we do not need a dynamics model and thus the training is simpler. **Cond** is a Transformer (Vaswani et al., 2017)-based demonstration-conditioned policy depicted in Figure 5. It digests a state-action demonstration of a task to perform and the observation history of an agent, and outputs the next action to take. It only has the alignment step due to its structure as discussed in Section 4.2. See Appendix B.5 for more comparisons and Appendix C.4 for more details of the training and implementation.

## 5.3 ALIGNMENT EVALUATION

**Quantitative Evaluation**  We evaluate the quality of the alignment in Maze2D, where we know the ground-truth state correspondence between domains. We expect that the corresponding states are mapped to the same latent representation. We calculate a prediction error of states in the source domain from the corresponding states in the target domain and use it as a metric of the alignment. Here we additionally train the state decoder $\phi_x^{-1}(s_z)$ to calculate the metric. Note that this metric is advantageous to GAMA since it has a direct mapping function of states. The scores in Table 1 show that, while Ours achieves similar scores to GAMA in Maze2D-O, our methods fail to accurately recover the state in the other cases. It is not only because our methods do not learn direct mappings between domains, but also because the model discards part of the information on the original state when projected into the domain-invariant space. As long as latent states can use the same common policy $\pi_z$, it can be valid to map the corresponding states to different positions, and hence a worse alignment score does not necessarily mean worse transfer performance. We examine how it affects the final transfer performance in Section 5.4. Regarding the performance of CCA, it is close to the perfect score in some settings since the observation shift is linear in Maze2D, whereas CCA fails in other cases due to the lack of explicit temporal alignment and the variations in the starting positions.

**Qualitative Evaluation**  We visualize the distributions of the learned latent state space using t-SNE (Van der Maaten & Hinton, 2008). As shown in Figure 7, Ours successfully aligns most pairs in successful cases in Maze2D-OA. In the more challenging environment, Maze-MR, the latent states

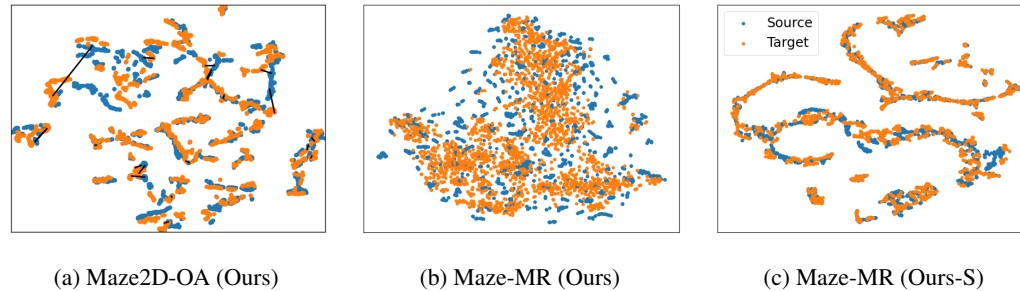

(a) Maze2D-OA (Ours)            (b) Maze-MR (Ours)            (c) Maze-MR (Ours-S)

Figure 7: Comparison of distributions of latent states $s_z$ in the medium maze visualized by t-SNE (Van der Maaten & Hinton, 2008). Similar to Figure 4, we sample corresponding states from domains and encode them into the learned feature space. Black lines connect 20 corresponding state pairs in each figure. In Maze-MR, we do not show the lines since the ideal alignment is not available.

Table 2: Success rates of target tasks. Results are averaged over nine seeds with three fixed goals.

| Task | Ours | Ours-S | GAMA | CCA | IfO |
|---|---|---|---|---|---|
| Maze2D-O (umaze) | $0.76 \pm 0.33$ | $0.42 \pm 0.41$ | $0.63 \pm 0.35$ | $0.45 \pm 0.35$ | $0.06 \pm 0.11$ |
| Maze2D-O (medium) | $0.97 \pm 0.07$ | $0.54 \pm 0.40$ | $0.77 \pm 0.24$ | $0.10 \pm 0.14$ | $0.00 \pm 0.01$ |
| Maze2D-OA (umaze) | $0.40 \pm 0.34$ | $0.36 \pm 0.36$ | $0.77 \pm 0.21$ | $0.45 \pm 0.35$ | $0.02 \pm 0.04$ |
| Maze2D-OA (medium) | $0.61 \pm 0.32$ | $0.72 \pm 0.29$ | $0.61 \pm 0.42$ | $0.10 \pm 0.14$ | $0.03 \pm 0.08$ |
| Maze-D (umaze) | $0.26 \pm 0.26$ | $0.50 \pm 0.41$ | $0.03 \pm 0.05$ | $1.00 \pm 0.00$ | $0.13 \pm 0.10$ |
| Maze-D (medium) | $0.36 \pm 0.29$ | $0.72 \pm 0.27$ | $0.01 \pm 0.03$ | $0.48 \pm 0.35$ | $0.00 \pm 0.01$ |
| Maze-MR (umaze) | $0.20 \pm 0.26$ | $0.42 \pm 0.32$ | $0.00 \pm 0.00$ | $0.03 \pm 0.07$ | $0.00 \pm 0.00$ |
| Maze-MR (medium) | $0.39 \pm 0.17$ | $0.48 \pm 0.24$ | $0.00 \pm 0.00$ | $0.00 \pm 0.00$ | $0.00 \pm 0.00$ |

for the target domain (i.e. Ant) have a broader distribution than the source domain. If we supply domain-specific state information $s$ to the decoder in Ours-S, the alignment is significantly improved since the encoder can focus more on domain-agnostic information for the action prediction.

## 5.4 Cross-Domain Transfer Performance

We measure the performance of the cross-domain transfer by the success rate of a target task. We choose a single goal for the target task and use the other goals as proxy tasks for the alignment phase. The values are averaged over three different target tasks. Table 2 summarizes the scores of methods that employ representation alignment in various settings. Note that CCA and IfO use RL for the adaptation and does not perform zero-shot transfer. In Maze2D, our methods and GAMA achieve better performance than CCA and IfO. Interestingly, our methods show comparable performance to GAMA even though the alignment score is worse than that of GAMA. This shows that aligning the latent representation is sufficient for transferring knowledge, instead of learning a direct mapping between domains. In Maze-D and Maze-MR, GAMA fails to transfer the policy, while our methods achieve a non-zero success rate consistently. Since it is difficult to find an exact correspondence between states with different complexities, direct state mapping and action mapping in GAMA are hard to acquire. Our methods instead reduce each MDP to the common one and thus we do not have to learn the direct relationship between the source domain and target domain. However, the performance of Ours is limited since the decoder cannot get sufficient domain-specific information for domain-specific action prediction. Using raw states $s$ directly fed to the decoder, Ours-S achieves the better performance with the improved alignment shown in Figure 7c. Regarding the performance drop of Ours-S in Maze2D-O, we give some explanation in Appendix B.4. We also compare our methods to Cond in Maze2D environments. The results in Figure 8 show that Cond struggles to adapt to the target task since the lack of the adaptation phase requires the model to perform zero-shot generalization to unseen demonstrations of a target task. This supports our claim in Section 4.2 that adaptation through updating a domain-agnostic policy on a shared representation space is effective for cross-domain transfer.

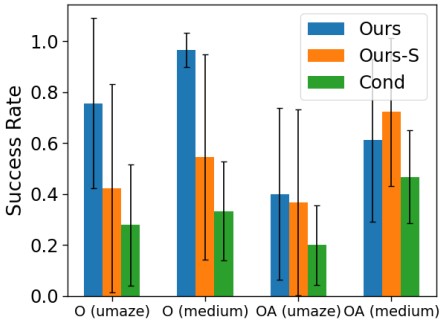 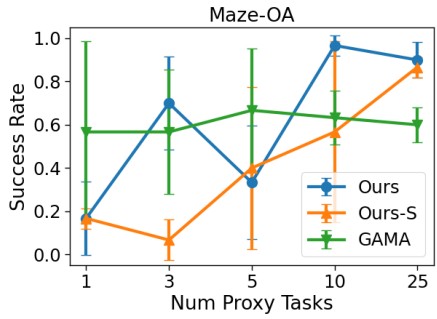

Figure 8: Success rates of our methods and a conditional model in Maze2D. Our models achieve better performance consistently because it can be trained only with source domain demonstrations. The error bars show the standard deviations.

Figure 9: The number of proxy tasks vs success rate in Maze2D-OA. The scores are averaged over three runs with a fixed goal. The error bars show the standard deviations. More results are available in Appendix B.1.

Table 3: Success rates of Ours-S in Maze2D (medium) with various coefficients $\lambda$ for the adversarial term. Here we used a fixed single goal. The scores are averaged over three runs.

| Task | $\lambda = 0$ | $\lambda = 0.1$ | $\lambda = 0.5$ | $\lambda = 2.0$ | $\lambda = 10$ |
|------|------|------|------|------|------|
| Maze2D-O | $1.00 \pm 0.00$ | $0.67 \pm 0.40$ | $0.67 \pm 0.40$ | $0.07 \pm 0.09$ | $0.53 \pm 0.33$ |
| Maze2D-OA | $0.93 \pm 0.09$ | $0.93 \pm 0.05$ | $0.87 \pm 0.05$ | $0.90 \pm 0.08$ | $0.67 \pm 0.34$ |
| Maze-D | $0.13 \pm 0.08$ | $0.09 \pm 0.09$ | $0.72 \pm 0.27$ | $0.89 \pm 0.09$ | $0.55 \pm 0.35$ |
| Maze-MR | $0.09 \pm 0.03$ | $0.61 \pm 0.24$ | $0.48 \pm 0.25$ | $0.26 \pm 0.21$ | $0.47 \pm 0.10$ |

## 5.5 ABLATION STUDIES

**Alignment Complexity** We measure the transfer performance, varying two aspects of the dataset for alignment: the number of trajectories and the number of proxy tasks. We present plots of the number of proxy tasks for Maze2D-OA in Figure 9 and more in Appendix B.1. These plots show that a decrease in the number of demonstrations or proxy tasks leads to a decrease in performance.

**Sensitivity to $\lambda$** Adversarial training is notorious for its difficulty in balancing the training of multiple functions: the encoder and the discriminator in our case. To evaluate how sensitive our method is to the adversarial coefficient $\lambda$, we measure the success rates of Ours-S with various values of $\lambda$. The results are shown in Table 3. Surprisingly, in Maze2D-O and OA, the transfer is successful without the adversarial term. In the other environments, on the other hand, adversarial training is necessary to align the representations, and we have to choose the correct value of $\lambda$.

## 6 CONCLUSION

In this work, we present a novel method to learn a domain-invariant policy in a common feature space for cross-domain policy transfer. Our experimental results show that our method achieves comparable performance to the prior methods in environments with domain shifts on the same agent and dynamics. Our method is especially effective for transfer between domains with different complexities, where no exact correspondence exists.

The main limitation of our work is the instability of the training due to adversarial training. Introducing techniques for more stable training of GANs could enhance the performance of our method. Alternatively, some self-supervised objectives can give more signals for the alignment and consequently stabilize or replace the adversarial training. Besides, relaxing the requirements on the dataset is a promising direction for future work. If we do not require expert actions or task ID in a dataset for the alignment, we can utilize prevalent, less-structured datasets including videos to scale the training. We hope our work provides some suggestions to researchers who will work on the development of a domain-free policy, an abstract policy that can be applied to any domain in a zero-shot manner.

REPRODUCIBILITY STATEMENT

We provide detailed descriptions of our experimental setups and implementations in Section 5.1 and Appendix C. These sections contain the key hyperparameters, links to the datasets, modified parts of the existing environments, dataset size, duration of the training, and other necessary information for reproduction. The training time does not exceed about five hours with a single GPU. We also release our codebase and created datasets as supplementary materials.

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

## A  DERIVATION AND INTERPRETATION OF LEARNING OBJECTIVE

Our objective in (1) is to maximize the entropy of domain prediction given a latent state

$$\max_\phi H(D|\mathcal{S}_z) = -\max_\phi \mathbb{E}_{d\sim p(D|\mathcal{S}_z)}\left[p(d|s_z)\right]$$

$$= \min_\phi \mathbb{E}_{d\sim p(D|\mathcal{S}_z)}\left[p(d|\phi_d(s_d))\right],$$

where $s_d$ is sampled from the dataset $\mathcal{D}$. Here we cannot directly evaluate $p(d|s_z)$. We introduce the variational approximator $q(d|s_z)$, which can be seen as a domain discriminator. Then we have

$$\min_\phi \mathbb{E}_{d\sim p(D|\mathcal{S}_z)}\left[q(d|\phi(s_d)) + p(d|\phi(s_d)) - q(d|\phi(s_d))\right]$$

$$= \min_\phi \mathbb{E}_{d\sim p(D|\mathcal{S}_z)}\left[q(d|\phi(s_d))\right] + D_{\text{KL}}\left[p(d|\phi(s_d))\|q(d|\phi(s_d))\right]$$

$$= \min_\phi \mathbb{E}_{(s_d,d)\sim\mathcal{D}}\left[q(d|\phi(s_d))\right] + D_{\text{KL}}\left[p(d|\phi(s_d))\|q(d|\phi(s_d))\right].$$

In the second equation, we remove the necessity of the sampling from the posterior following the discussion similar to Lemma 5.1 in Chen et al. (2016).

Since we would like to minimize this objective instead of maximizing it as done in the well-known evidence lower-bound (ELBO) objective, we cannot take a lower bound by skipping the second term. In our case, we can consider that the training of the discriminator $q$ toward correct domain classification is a minimization of the KL divergence. Given the sufficient quality of the approximation, the KL term should be small, so the encoder can focus on optimizing the first term, which is our final objective in (2).

## B  ADDITIONAL RESULTS AND DISCUSSIONS

### B.1  ADDTIONAL RESULT OF ALIGNMENT COMPLEXITY

Figure 10 and Figure 11 show the alignment complexity in Maze2D-O and Maze2D-OA with respect to the number of demonstrations and the number of proxy tasks. As mentioned in Section 5.5, an increase in the number of demonstrations or proxy tasks improves the performance.

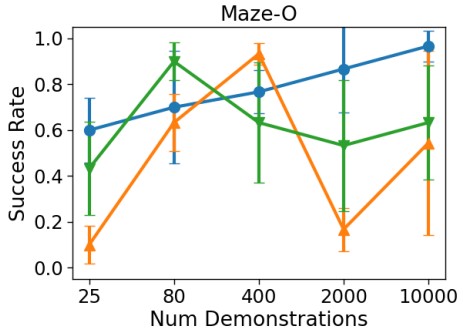 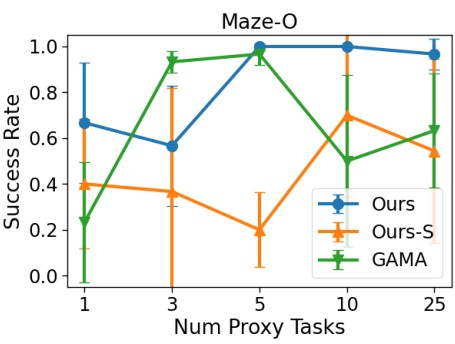

(a) The number of demonstrations vs success rate

(b) The number of proxy tasks vs success rate

Figure 10: Alignment complexity in Maze2D-O. Here we used a fixed single goal. The scores are averaged over three runs. The error bars show the standard deviations.

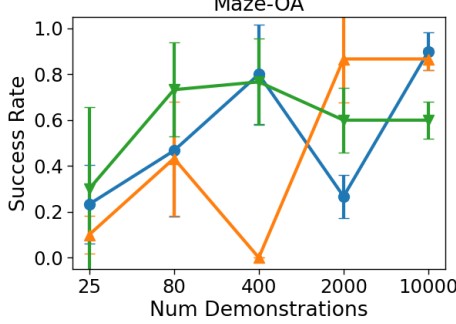 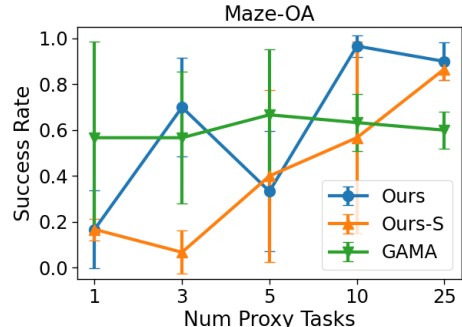

(a) The number of demonstrations vs success rate

(b) The number of proxy tasks vs success rate

Figure 11: Alignment complexity in Maze2D-OA. The scores are averaged over three runs with a fixed goal. The error bars show the standard deviations.

Table 4: Success rates in Maze2D (medium) with and without behavioral cloning. Results are averaged over nine seeds with three fixed goals.

| Task | BC + adversarial | adversarial only |
|---|---|---|
| Maze2D-O (Ours) | $0.97 \pm 0.07$ | $0.03 \pm 0.05$ |
| Maze2D-OA (Ours) | $0.61 \pm 0.32$ | $0.03 \pm 0.07$ |
| Maze2D-O (Ours-S) | $0.54 \pm 0.40$ | $0.03 \pm 0.04$ |
| Maze2D-OA (Ours-S) | $0.72 \pm 0.29$ | $0.00 \pm 0.00$ |

Table 5: Success rates of target tasks, with and without task conditioning of the discriminator. Results are averaged over nine seeds with three fixed goals.

| Task | Ours-S | Ours-S (w/o task) | GAMA | GAMA (w/o task) |
|---|---|---|---|---|
| Maze2D-O (umaze) | $0.42 \pm 0.41$ | $0.46 \pm 0.42$ | $0.63 \pm 0.35$ | $0.69 \pm 0.23$ |
| Maze2D-O (medium) | $0.54 \pm 0.40$ | $0.27 \pm 0.27$ | $0.77 \pm 0.24$ | $0.78 \pm 0.16$ |
| Maze2D-OA (umaze) | $0.36 \pm 0.36$ | $0.47 \pm 0.38$ | $0.77 \pm 0.21$ | $0.38 \pm 0.35$ |
| Maze2D-OA (medium) | $0.72 \pm 0.29$ | $0.67 \pm 0.24$ | $0.61 \pm 0.42$ | $0.48 \pm 0.35$ |

## B.2 SUCCESS RATES WITHOUT BEHAVIORAL CLONING

Although we align the representation through behavioral cloning and the adversarial objective jointly, one might think that we can first align the representations with the adversarial term only and directly learn the common policy in the learned representation space. To measure how much behavioral cloning helps the alignment, we measure the success rates when the alignment is only done with the adversarial term. Table 4 shows that behavioral cloning encourages the model to learn the representation space that incorporates the closeness of states in terms of the actions to perform in proxy tasks. The adversarial term is insufficient for the alignment because it only matches the distributions of latent states of the source domain and the target domain. It implies that we could get a better alignment if we impose appropriate additional constraints on the representations.

## B.3 EFFECT OF TASK CONDITIONING

We employ a task-level alignment signal by feeding task ID $k$ to the discriminator. It encourages the model to align not only the entire latent state distributions but also the ones in every single task. We measure the success rates of target tasks with and without task conditioning of the domain discriminator. The results in Table 5 show that the task-level alignment improves the performance of both our method and GAMA in some cases. Based on this result, we use task conditioning by default in our experiments.

## B.4 PERFORMANCE DROP OF OURS-S IN MAZE2D-O

As shown in Table 2, Ours-S, which takes raw states $s$ as input of the action decoder $\psi^{-1}(a_z)$, can sometimes show worse performance than Ours. Since the decoder can directly use domain-specific state information $s$ for the action prediction, the common policy can ignore and discard the information of the latent state $s_z$ and only pass latent action that is equivalent to a task ID $k$. In this case, the transfer will fail because the decoder will encounter an unknown task ID of a target task passed from the common policy at inference time. We conjecture that Ours-S fails when it falls into this failure mode. It can possibly be solved by controlling the amount and the content of the information provided to the decoder from a raw state $s$.

## B.5 ADDITIONAL BASELINE

As an additional baseline, we measure the transfer performance of XIRL (Zakka et al., 2022). **XIRL** first learns a cross-domain state representation with TCC (Dwibedi et al., 2019) that captures task progression, and utilizes it to shape the reward for performing RL in the adaptation. It thus does not perform zero-shot transfer. The results are shown in Table 6. The results show that our methods consistently outperform XIRL without interacting with the environment in the adaptation phase.

Table 6: Success rates of target tasks (Ours, Ours-S, and XIRL). The results of the other methods are shown in Table 2. The values are averaged over nine seeds with three fixed goals.

| Task | Ours | Ours-S | XIRL |
|------|------|--------|------|
| Maze2D-O (umaze) | $0.76 \pm 0.33$ | $0.42 \pm 0.41$ | $0.30 \pm 0.33$ |
| Maze2D-O (medium) | $0.97 \pm 0.07$ | $0.54 \pm 0.40$ | $0.03 \pm 0.09$ |
| Maze2D-OA (umaze) | $0.40 \pm 0.34$ | $0.36 \pm 0.36$ | $0.30 \pm 0.33$ |
| Maze2D-OA (medium) | $0.61 \pm 0.32$ | $0.72 \pm 0.29$ | $0.03 \pm 0.09$ |

Table 7: Success rates of the target task in the robotic manipulation enviroment (Ours, Ours-S, and GAMA). The values are averaged over nine seeds with three fixed target positions.

| Ours | Ours-S ($\lambda = 0.0$) | Ours-S ($\lambda = 0.5$) | GAMA |
|------|------|------|------|
| $0.00 \pm 0.00$ | $0.61 \pm 0.33$ | $0.86 \pm 0.22$ | $0.18 \pm 0.27$ |

Note that Maze2D-O is equivalent to Maze2D-OA for XIRL and we only measure the performance in Maze2D-O since XIRL learns the state correspondence only with states.

### B.6 PERFORMANCE IN MANIPULATION TASK

We additionally evaluate our methods in a robotic manipulation task in robosuite (Zhu et al., 2020) framework (Figure 12). We use the Block Lifting task, where the robot has to pick up a block and lift it to a certain height. A task is defined as the position of the object to lift. We set nine initial locations on the table. We use one for the target task and the rest for proxy tasks for aligning representations. To evaluate the transfer between different robots, we use Sawyer for the source domain and UR5e for the target domain. Sawyer is a 7-DoF robot and UR5e is a 6-DoF robot, hence these robots have different observation spaces and dynamics. The results in Table 7 demonstrate that Ours-S outperforms GAMA. As in Maze-D and Maze-MR, GAMA shows suboptimal performance since there is no good correspondence across robots with different DoFs and grippers. Ours does not perform well in this environment either, probably because the domain-specific state input to the decoder is highly necessary for the accurate control in this environment. Ours-S avoids these issues and successfully performs cross-robot zero-shot transfer. We also confirm that the adversarial term boosts the performance of our method.

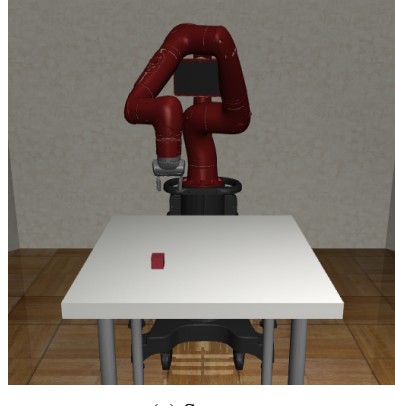
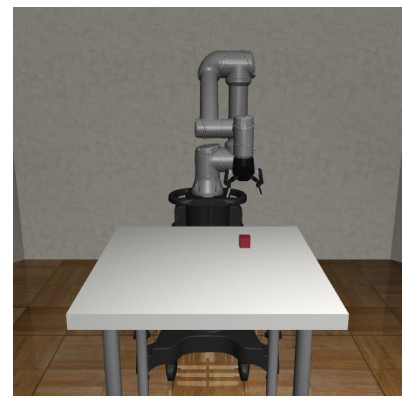

(a) Sawyer

(b) UR5e

Figure 12: The pictures of the robotic manipulation environments.

## C  EXPERIMENT DETAILS

### C.1  ENVIRONMENTS

All Maze environments used in this paper are based on D4RL (Fu et al., 2020). In Maze environments, two types of agents perform actions: Point and Ant. The Point agent has a state space of four dimensions and an action space of two dimensions. The states consist of positions and velocities for the x-axis and y-axis, while the actions consist of the force to be applied for each direction. The Ant agent has a state space of 29 dimensions and an action space of eight dimensions. The states consist of the position and velocity of the body, and the joint angles and angular velocities of the four legs. The actions consist of the force to be applied to each joint.

In Maze-D and Maze-MR, we use an environment with different dynamics from the original environment for the Point agent. We removed the inertia from the source domain and thus changes in position depend only on the action at the step. In Maze-D, the task of the Point agent in the target domain is to learn from the trajectories of an agent in the source environment that does not have inertia. In Maze-MR, the task of the Ant agent is to learn from the Point agent in the environment with the simpler dynamics. The input positions of x and y coordinates of the Point agent were reversed from those of the Ant agent to prevent leaking information without representation alignment.

For the manipulation task, we use robosuite framework (Zhu et al., 2020). We use Sawyer for the source domain and UR5e for the target to test cross-robot transfer (Figure 12). We choose Block Lifting task, where the robot has to pick up a block and lift it to a certain height. A task is defined as the position of the object to lift. We set nine initial locations on the table. We use one for the target task and the rest for proxy tasks for aligning representations. The initial pose of the robot is randomized. The observation space consists of the position or angular information and the velocity of each joint and the end effector. Sawyer and UR5e have state spaces of 32 dimensions and 37 dimensions, respectively. The input positions of state elements of Sawyer were reversed for the same reason as Maze-MR. Both robots are controlled by delta values of the 3D position of the end effector and 1D gripper state.

### C.2  DATASETS

As described in Section 3, the dataset contains state action sequences of expert demonstrations with various goals (i.e., multiple tasks). For the Maze2D and Maze-D, we provide about 10k trajectories of six tasks for the umaze, and about 10k trajectories of 25 tasks for the medium maze unless explicitly mentioned in the ablation study. We downloaded the expert demonstrations from `http://rail.eecs.berkeley.edu/datasets/offline_rl/maze2d/` (maze2d-umaze-sparse-v1, maze2d-medium-sparse-v1). For Maze-MR, we provide about 10k trajectories of six tasks for the umaze and about 5k trajectories of 25 for the medium maze. When we created expert trajectories for the AntMaze and simpler Maze2D environement without inertia in Maze-D, we used PPO (Schulman et al., 2017) from stable-baselines3 (Raffin et al., 2021) to train agents to move only one square up, down, left, or right. We then composed entire demonstrations by solving the maze with BFS and giving the agent the direction of the next square. For the robotic manipulation environment, we provided 600 trajectories for each task ID. We collected the expert demonstrations by a scripted policy based on the object position and the gripper pose.

### C.3  ARCHITECTURE AND TRANING DETAILS OF OUR METHOD

Our policy is a simple multilayer perceptron. In the experiments of Maze2D, the state encoder, the common policy, and the decoder have three, five, and three hidden layers with 256 units and ReLU activations. Only the last layer of the decoder has Tanh activation. The dimension of latent representations is also 256. We optimized our objective with Adam optimizer (Kingma & Ba, 2015). We set the learning rate to 1e-4 and the batch size to 256, and trained the model for 20 epochs. In Maze-D, we used the same number of layers as in Maze2D, while, in Maze-MR, we used four, six, and four hidden layers for the encoder, the policy, and the decoder, respectively. In Maze-D and Maze-MR, we used Tanh activation for the last layer of each component, and Mish activation for the other layers. The number of units and the dimension of the latent representation were set to 512. We set the learning rate to 2e-4 and the batch size to 512, and trained the model for 40 epochs.

### C.4 BASELINES

For GAMA, we re-implemented the algorithm referring to the original paper and an official implementation (https://github.com/ermongroup/dail). When we found a few differences between the paper and the implementation, we followed the description in the paper. We swept the adversarial coefficient from 0.01 to 10, the learning rate from 1e-4 to 1e-3, and used 0.5 and 1e-4, respectively. For CCA, we used the trajectories of the proxy tasks to learn linear state correspondence mappings. We first padded trajectories to the same length and created a single sequence of trajectories for each domain by concatenating them sorted by task ID. The order of trajecories were randomly shuffled within a task ID. In the adaptation phase, a policy is trained by reinforcement learning using an auxiliary reward function $r(s_y^{(t)})$ defined as follows:

$$r(s_y^{(t)}) = -\frac{1}{|\mathcal{D}_x|} \sum_{\tau \in \mathcal{D}_x} \|g(s_y^{(t)}) - f(s_{x,\tau}^{(t)})\|_2^2,$$

where $f(s_x)$ is a learned mapping function from the source (expert) domain to the learned latent space, $g(s_y)$ is its counterpart for the target (agent) domain, $\mathcal{D}_x$ is the expert trajectories of a target task in the source domain, $s_y^{(t)}$ is a state of the target agent at time step $t$, and $s_{x,\tau}^{(t)}$ is a target state in the source domain at time step $t$ in a sampled expert trajectory $\tau$. If the state correspondence is learned sufficiently, $r(s_y^{(t)})$ would be a reward for tracking features of expert trajectories in the source domain. We used the Deep Deterministic Policy Gradient (DDPG) (Lillicrap et al., 2015) algorithm for this step and trained MLP policy for 200k environmental steps. For IfO, we trained a context translation model which translates expert demonstrations in the source domain into the ones in the target domain. We randomly chose pairs of demonstrations performing the same task and trained the model to perform translation between the corresponding states found by dynamic time warping (DTW) (Müller, 2007). In the adaptation phase, we used DDPG to train the policy in the target domain for 200k environmental steps. As in CCA, the reward is defined to track the features of the expert demonstrations in the source domain.

XIRL uses the domain-invariant feature acquired by TCC (Dwibedi et al., 2019) to shape the reward for the adaptation. Given the demonstration in the source domain, the reward is calculated as follows:

$$r(s) = -\frac{1}{\kappa} \left\| \phi(s) - \frac{1}{N} \sum_i \phi(s_{T_i}) \right\|_2,$$

where $\phi(s)$ is the state embedding, $T_i$ is the length of demonstration $i$, $N$ is the number of demonstrations. Intuitively, it is a distance of a current state from the averaged goal in the feature space. $\kappa$ is a scale parameter and we set it to 10% of the distance of the first state from the goal state. We tried the 1%, 10%, and 100% and chose the best one for our setting. The policy was trained with SAC (Haarnoja et al., 2018) as in the original paper. We trained the policy for 800k environmental steps with learning rate 3e-4. TCC was trained with the classifier-based loss, which performed best in our expeiriments. For more details, please refer to the original paper.

For the demonstration-conditioned model (Cond), we used a Transformer (Vaswani et al., 2017)-based architecture to process sequences of observations and actions (Figure 5). We fed demonstrations and observation history as they were without thining-out timesteps. The maximum sequence length was 250 and 400 for the umaze and medium maze, respectively. The model had three encoder layers and three decoder layers with 256 units and eight heads for each layer. The dropout rate was set to 0.2. The activation function was ReLU and it was applied after the normalization. We set the batch size to 32, the learning rate to 1e-3, and trained the model for 100 epochs. We confirmed that the error of behavioral cloning was going down to the similar value observed in the training of our method and GAMA.

