# OpenReview forum: "Learning a Domain-Agnostic Policy through Adversarial Representation Matching for Cross-Domain Policy Transfer"
_ICLR.cc/2023/Conference — Submitted to ICLR 2023_

### Official Review · Reviewer_rLRK · 2022-10-25

**Confidence:** 2
**Correctness:** 3
**Technical Novelty And Significance:** 2
**Empirical Novelty And Significance:** 2
**Recommendation:** 6

**Clarity, Quality, Novelty And Reproducibility:**

Clarity
The paper is easy to understand, and the quality looks good.



**Strength And Weaknesses:**

Strength
The paper proposes to use approach similar to GAN that aims to acquire domain-invariant feature space and a common policy on it. The author provided extensive ablation study, and compared the model with state of the art models, and showed the new model has better performance.
Weaknesses
The training process seems quite complicated. Combined with the instability of adversarial training, the model can be difficult to tune.

**Summary Of The Paper:**

This paper provides a novel method of cross-domain transfer with an unaligned dataset, and the proposed approach aims to acquire domain-invariant feature space and a common policy on it. And the experiments with various domain shifts show that the proposed method achieves better performance than existing methods in cross-dynamics.

**Summary Of The Review:**

The paper is well written and provided extensive experiments to show the proposed method works better than baseline models.

---

> ### Author Response · Authors · 2022-11-18
> **Response to Reviewer rLRK**
>
> Thank you for your time and helpful feedback on our paper.
> > The training process seems quite complicated. Combined with the instability of adversarial training, the model can be difficult to tune.
>
> We believe our training procedure is not quite complicated. In the alignment phase, we simply train all the layers
> jointly with behavioral cloning and adversarial training. In the adaptation phase, we freeze the encoder and decoder and update the parameters of the common policy. It should not make the learning in the adaptation phase difficult, given successful representation alignment.
> However, as you pointed out, the instability of adversarial learning can lead to unstable performance of the resulting policies. As we mentioned in Conclusion, it is one of the limitations and important future work of our paper.

---

### Official Review · Reviewer_X2sb · 2022-10-25

**Confidence:** 4
**Clarity, Quality, Novelty And Reproducibility:** 1. Clarity and quality
**Correctness:** 3
**Technical Novelty And Significance:** 2
**Empirical Novelty And Significance:** 2
**Recommendation:** 5

**Strength And Weaknesses:**

Strength:
1. It is a simple yet reasonable idea for cross-domain policy transfer by aligning and training a policy in a common latent representation space shared by different domains.
2. The authors provide both qualitative and quantitative results to show the performance of adversarial alignment.
3. In general, the paper is well-written and easy to follow.

Weaknesses:
1. My main concern is the novelty of the proposed approach, which seems very similar to GAMA. Both of them use an adversarial objective for state alignment in different domains so that the policy can be directly transferred to the target domain. The major difference is that in this paper, the states are aligned in a common latent space instead of a direct matching.
2. If I understand correctly, the action $a_z$ from the common policy may only contain compact information. Therefore, if action $a_d$ in the target domain is more complicated than the action in the latent space, only using $a_z$ and the domain ID might not be able to decode $a_d$. For example, $a_z$ may represent a high-level behavior such as “turn left”. However, for a robot to execute this action, it may take multiple steps based on its current domain-specific states (e.g., the pose of its own torso). In other words, the method might only work when the source and target domains have a strong relationship, because the policy in the target domain is completely determined by the common state, and does not depend on any domain-specific skills.
3. All experiments are conducted in the Maze environments in the D4RL environments, which is inadequate to support the effectiveness of the model in general control scenarios. Therefore, I would suggest validating the model on other benchmarks with a wide variety of dynamics. For a fair comparison, the authors may use the original environments of other baselines, e.g., those used in the GAMA paper.
4. Except for GAMA, the compared approaches are somewhat "out-of-date". Although I understand that this paper is focused on performing policy transfer between offline datasets, without interacting with various environments, it would be nice if the authors could include stronger baseline methods for model comparison, especially the offline RL approaches that show impressive performance on the D4RL dataset.
5. A typo: “deomnstrations” in Appendix C.2.


**Summary Of The Paper:**

This paper presents a novel method to perform zero-shot cross-domain transfer by learning a domain-agnostic policy in the source domain that can be directly applied to the target domain. To this end, specific training methods (technical contritions) include:
1. Align representations of states and actions from different domains to a common latent space through adversarial training.
2. The common policy is trained to adapt to the target task in the source domain.
3. Directly apply the common policy to the target domain based on the aligned representations.

The effectiveness of the proposed approach is verified with four Maze environments in the D4RL benchmark, which is shown to outperform other baselines especially when there are sufficient proxy tasks.

**Summary Of The Review:**

Overall, I think the paper presents a simple but effective cross-domain policy transfer method. However, I am inclined to reject this paper given its technical novelty compared with GAMA and the insufficient experimental results.

---

> ### Author Response · Authors · 2022-11-18
> **Response to Reviewer X2sb**
>
> Thank you for reviewing our paper and providing detailed feedback.
> > My main concern is the novelty of the proposed approach, which seems very similar to GAMA.
>
> > The major difference is that in this paper, the states are aligned in a common latent space instead of a direct
> matching.
>
> It is a correct understanding of our paper. Our point is that this seemingly small difference can have a significant impact on the transfer performance especially when there is no explicit mapping between domains that GAMA tries to obtain.
> In fact, our approach shows better performance than GAMA in such environments even though our approach is even simpler than GAMA.
>
> > If I understand correctly, the action $a_z$ from the common policy may only contain compact information. Therefore, if action $a_d$ in the target domain is more complicated than the action in the latent space, only using $a_z$ and the domain ID might not be able to decode $a_d$.
>
> It is a very good point to discuss. Thank you for pointing it out. Since we maximally remove domain-specific
> information from the latent representations, it can have a large negative impact on the action prediction that needs domain-specific content. We can compensate for the lack of information by feeding raw states to the action decoder.
> This approach also can lead to an undesirable situation where the common policy only passes the task ID and discards the latent states from the encoder. Then the transfer will fail since the decoder will receive an unknown vector that is equivalent to the task ID of a target task.
> To investigate which case is more likely to happen, we added this approach, named Ours-S, as a variant of Ours in the revision. According to the results, the variant works well especially when one domain is more complex than the other, and the aforementioned problem rarely happened in our experiments. We hope it will partly address your concern as well.
>
> > All experiments are conducted in the Maze environments in the D4RL environments, which is inadequate to support the effectiveness of the model in general control scenarios.
>
> We have added a robotic manipulation task and compared the performance of our method and GAMA in Appendix B.6.
> Although our method did not perform well without states fed into the decoder, Ours-S significantly outperformed GAMA in the experiment.
>
> > Except for GAMA, the compared approaches are somewhat ”out-of-date”.
>
> We have added baselines that are not too ”out-of-date” but require interaction with the environment: IfO [1] and XIRL [2]. Neither IfO nor XIRL performed well in our experiments even though they can interact with the environment in the adaptation.
>
> > A typo: “deomnstrations” in Appendix C.2.
>
> We fixed it. Thank you.
>
> References
> [1] Liu et al. Imitation from observation: Learning to imitate behaviors from raw video via context translation. ICRA, 2018.
> [2] Zakka et al. XIRL: Cross-embodiment inverse reinforcement learning. CoRL, 2021.

---

> > ### Comment · Reviewer_X2sb · 2022-12-11
> > **Appreciate the responses and have increased my score**
> >
> > I would like to thank the authors for their responses and extra results, which have solved most of my concerns about the experiments, and so I decided to sightly increase my score (3->5).
> >
> > But I still think the novelty part does not suffice (especially the overall idea compared with GAMA).

---

### Official Review · Reviewer_QGQp · 2022-10-26

**Confidence:** 3
**Correctness:** 3
**Technical Novelty And Significance:** 3
**Empirical Novelty And Significance:** 3
**Recommendation:** 6

**Clarity, Quality, Novelty And Reproducibility:**

The paper could benefit form some further improvements in the writing, especially on the first page, i.e. pages 1-3. It is a good paper though, with fair novel and original components, Code has been provided hence it is easily reproducible.

**Strength And Weaknesses:**

Strengths:

a) Relatively simple method, which can be extended and further improved relatively easily, especially the adversarial training part, and of course the learning part.
b) Competitive results show promising avenues for expanding on other settings and in other environments
c) Visualisations of the latent space provide some further understanding of the underlying processes involved.

Weaknesses:

a) Background work is rather blunt. The last paragraph on page 2 is not well written. Methods' names are somewhat akwardly provided alongside the in-text citation.
b) I am not sure that the background covers all recent methods, but because I am not very well aligned with this area I am not very firm on it.
c) Further to b) more comparisons could have been added to the main text
d) Methods are not very well described in the main paper - appendix provides further context though so it somewhat compensates. It buffles me though that the method works relatively well, given what the methods employed

**Summary Of The Paper:**

The proposed method is as simple as aligning representations in the latent space to learn a domain-invariant policy. Conceptually that makes sense as it can generalise better in the presence of small perturbations and changes in an environment. This allows to do some domain adaptation with zero-shot learning whereby everything learnt in the source domain (environment Χ) can be transferred into the target domain (environment Υ). The results presented are fairly comprehensive and the visualisations alike, but the description of the method is not very convincing, i.e. it seems to work surprisingly well. The authors have provided the code and the corresponding datasets though.

**Summary Of The Review:**

The paper provides adequate experiments and visualisations to justify the advantages of the proposed method. I am somewhat hesitant due to the method description seeming too straightforward and 'simple' (not in a bad way, in that I do not mean that complex is needed) to learn representations that are transferable and generalisable. Results prove me wrong though so am erring in caution.

---

> ### Author Response · Authors · 2022-11-18
> **Response to Reviewer QGQp**
>
> Thank you for your time and helpful feedback on our paper. We also appreciate your recognition of the simplicity of our
> approach.
>
> > Background work is rather blunt.
>
> In the revision, we reorganized the descriptions of previous studies on cross-domain policy transfer.
>
> > I am not sure that the background covers all recent methods,
>
> We did our best to cover the most recent methods that are particularly similar to our setting except for the concurrent
> work we found recently [1], which employs a similar strategy but cannot perform zero-shot transfer.
>
> > more comparisons could have been added to the main text
>
> We added two more recent methods, IfO [2] and XIRL [3], which allow interaction with the environment when adapting
> to a target task.
>
> > Methods are not very well described in the main paper
>
> > the description of the method is not very convincing
>
> The reason why our method worked well despite its simplicity is that behavioral cloning and adversarial training
> encourage cross-domain representation matching complementarily. Behavioral cloning seems to shape the structure of the
> distribution of each domain (Fig. 4a), and the adversarial term pulls the entire distributions close together so that they
> will be identical. This hypothesis is partly supported by our ablation studies in that the distributions can be aligned only
> with the behavioral cloning term in the ideal case (Table 3) whereas they cannot be aligned only with the adversarial
> term (Table 4 in Appendix). In the revision, we mentioned it more clearly in the last part of Section 4.1.
> Besides, the behavioral cloning objective is helpful to incorporate reward and dynamics information into the state
> representation. Some previous studies proved that it is helpful for the transfer. We conjecture that there is a relationship
> between the performance of our method and such a theoretical statement, especially when the performance is not bad despite the corresponding states are not identical in the latent space. A deeper theoretical analysis would be helpful in
> future work.
>
> References
> [1] Franzmeyer et al. Learn what matters: cross-domain imitation learning with task-relevant embeddings. NeurIPS,
> 2022.
> [2] Liu et al. Imitation from observation: Learning to imitate behaviors from raw video via context translation. ICRA,
> 2018.
> [3] Zakka et al. XIRL: Cross-embodiment inverse reinforcement learning. CoRL, 2021.

---

### Official Review · Reviewer_737K · 2022-11-17

**Confidence:** 3
**Correctness:** 3
**Technical Novelty And Significance:** 3
**Empirical Novelty And Significance:** 3
**Recommendation:** 5

**Clarity, Quality, Novelty And Reproducibility:**

Clarity: the paper is well-written and easy to follow.

Quality: it has a self-contained framework and proposed a classic solution to the problem.

Novelty: the novelty is limited, according to the weaknesses mentioned above.

Reproducibility: good. The source code is provided.

**Strength And Weaknesses:**

Strength:

The topic of cross-domain imitation learning is relatively new and has the potential to explore.

The framework of the paper (the common state and action space) is clearly stated and can inspire the following works.

The method introduces domain adaptation learning into behavior cloning, which is relatively new in the community.

Weakness:

Domain adversarial learning is a conventional method for domain adaptation when we need to learn domain-invariant representations. The paper directly uses domain adversarial learning to learn the common space of the state. It is not a very novel idea for me.

GAMA uses domain adversarial learning to learn the mapping of the state and the action between domains. The paper is more like a simple version of GAMA, as the domain discriminator of GAMA takes the triples (s, a, s') as input while the paper takes the state s as input. GAMA learns the mapping from the source state and action to the target while the paper learns a common space. Therefore, the contribution of the paper is incremental to GAMA.

Intuitively speaking, the adaptation phase might disturb the aligned space learned by the alignment phase. Because in the adaptation phase, the method will finetune the common policy for the source domain, it will cause inconsistency between the common policy and the decoder of the target domain.

The alignment scores in Table1 clearly do not favor the proposed method. If the paper would like to claim this alignment score is not a suitable metric, then it needs to propose some new metrics.

The experimental environments have no overlap with GAMA. Therefore, it is not convincing enough that the method is better than GAMA.

**Summary Of The Paper:**

The paper study the domain adaptation imitation learning. The paper proposes a method to map the state and the action of both domains into a common space. After learning this common space, the target agent can zero-shot perform the new task that only has source demonstrations. Specifically, the paper use domain adversarial learning to learn such common space of the state and the action.

**Summary Of The Review:**

Overall the paper clearly states the problem and the framework, which can inspire the readers. The following works can benefit from the view of the paper, which introduces the common state and action space.

However, the novelty of the work is limited. The paper follows GAMA and combines more ideas in the domain adversarial learning. Therefore, I will weakly reject this paper and hope the author can improve the idea and the experiments.

---

> ### Author Response · Authors · 2022-11-18
> **Response to Reviewer 737K**
>
> Thank you for your time and valuable feedback on our paper.
>
> > Domain adversarial learning is a conventional method for domain adaptation. ... It is not a very novel idea for me.
>
> > the contribution of the paper is incremental to GAMA.
>
> As you pointed out, using domain adversarial learning to obtain domain-invariant representation is not novel (proposed
> in [1]), and a few previous studies on cross-domain imitation use similar approaches in different settings. For example, [2]
> applies a domain confusion objective on a state representation of a value function, and [3] applies it to the intermediate
> feature of cross-domain state translation by CycleGAN. We apply it to the state representation in a policy trained with
> behavioral cloning and use it for zero-shot cross-domain transfer.
> The core difference between our approach and GAMA is that we learn a common space instead of direct cross-domain
> mapping. Surely, the difference is not large, but our claim is that this relatively small change has a critical effect on the
> transfer performance especially when the complexities of domains are very different such as in cross-morphology transfer.
> Our approach shows comparable or better performance than GAMA though it is even simpler than GAMA. We believe
> that the simplicity contributes to the extensibility for the following work as well.
>
> > Intuitively speaking, the adaptation phase might disturb the aligned space learned by the alignment phase. Because
> in the adaptation phase, the method will finetune the common policy for the source domain, it will cause inconsistency
> between the common policy and the decoder of the target domain.
>
> We assume that the alignment obtained in the alignment phase is good enough, that is, states that are equivalent for
> behavioral cloning are mapped to similar points in the common space. As you said, if the alignment is insufficient, we
> cannot combine a policy for the states in the source domain with the decoder for the target domain. However, since we
> fixed the encoder and the decoder in the adaptation phase, the common space and its relationship with raw states and
> actions are fixed during the adaptation phase. The potential inconsistency comes from the alignment phase, not from the
> adaptation phase.
>
> > The alignment scores in Table1 clearly do not favor the proposed method. If the paper would like to claim this
> alignment score is not a suitable metric, then it needs to propose some new metrics.
>
> Since one of our main claims is in the difference from GAMA, we put an emphasis on the comparison with it. In Table
> 1, we are not trying to show the strength of our method in terms of this metric, but to provide an analysis of how our
> method transfers knowledge. We adapted the same metric used in the GAMA paper and showed it did not retain all the
> information in the raw states. For the comparison with methods other than GAMA, which learn latent representation in
> their architecture, it would also be helpful if we had measured the quality in the latent space.
>
> > The experimental environments have no overlap with GAMA. Therefore, it is not convincing enough that the method
> is better than GAMA.
>
> We used environments with various domain shifts that include shifts in observations, actions, and dynamics in the
> Reacher environment in the GAMA paper. We also examined the performance in a more complex setting, Maze-MR, to
> highlight when our method works better. Although we did not use the same environment for the evaluation, we think it
> is a valid comparison that includes the situations tested in the original paper.
>
> References
> [1] Tzeng et al. Adversarial discriminative domain adaptation. CVPR, 2017
> [2] Zhang et al. Invariant causal prediction for block mdps. ICML, 2020
> [3] Raychaudhuri et al. Cross-domain imitation from observations. ICML, 2021.

---

> > ### Comment · Reviewer_737K · 2022-12-14
> > **Keep my score**
> >
> > I have read the author's response, which is very detailed. However, it didn't fully address my concerns, nor did it convince me.  I still think the weaknesses I mentioned in the weaknesses section are the limitations of the paper. So I keep my score.

---

### Author Response · Authors · 2022-11-18
**General Response: Summary of Paper Updates**

We thank all the reviewers for their time and helpful feedback. We have updated our paper based on the provided
comments. We highlight the following changes:
- We reorganized the related work on cross-domain policy transfer. We also added a few more methods of correspondence learning to the last paragraph of the related work section.
- We added explanations of how and why our method works well in the last paragraph of Section 4.1.
- We added results of a variant of our method, *Ours-S*, which additionally takes a raw state as an input of the decoder.
The state input to the decoder compensates the domain-specific information for the better action prediction that is
removed from the latent representations as much as possible. This can lead to an undesirable consequence where the
common policy learns to pass the task ID to the decoder as it is, ignoring the latent states from the encoder. In this
case, the transfer will fail since the decoder will encounter an unknown task ID passed from the common policy at
inference time.
However, in our additional experiments, it rarely happened and Ours-S worked well especially when
the complexities of domains are significantly different (Maze-D and Maze-MR). Since Ours-S still has the concern
we mentioned, and also it is slightly more complicated than our original method, we decided to keep the original
one instead of completely replacing it with Ours-S.
- We added more recent baselines: IfO [1] in the main text and XIRL [2] in Appendix B.5, although both of them
assume the interaction with the environment in the adaptation to the target task. Our method performed better
than these approaches in our experiments.
- We added an evaluation in a robotic manipulation task in Appendix B.6. Ours-S we added significantly outperformed
GAMA in the experiment.
- Considering the page limit, we shorten the description of task conditioning in the main text and moved it to
Appendix B.3.

References
[1] Liu et al. Imitation from observation: Learning to imitate behaviors from raw video via context translation. ICRA,
2018.
[2] Zakka et al. XIRL: Cross-embodiment inverse reinforcement learning. CoRL, 2021.

---

### Decision · Program_Chairs · 2023-01-20

**Decision:**

Reject

**Justification For Why Not Higher Score:**

Based on the reviews, as well as the AC's take on the paper, there is an agreement that the current method is too similar to GAMA.

**Justification For Why Not Lower Score:**

N/A

**Metareview: Summary, Strengths And Weaknesses:**

This paper studies a method for zero-shot cross-domain transfer by learning a domain-agnostic policy in the source domain that can be directly applied to the target domain. At the core of the approach is aligning representations of states and actions from different domains to a common latent space through adversarial training. The proposed approach is verified with four Maze environments in the D4RL benchmark. The approach is simple and reasonable, and the improvement in performance is beneficial. While the initial reviews are controversial, the author rebuttal addressed many of the concerns, such as extra results to solve most concerns about the experiments. However, the main concern held by the two negative reviewers is the novelty of the approach: The proposed approach is too similar to GAMA -- The difference from GAMA is not substantial enough to warrant another published paper. Authors are encouraged to further enhance their paper with a more pronounced algorithm.